# Three-Dimensional Combined Atrioventricular Coupling Index—A Novel Prognostic Marker in Dilated Cardiomyopathy

**DOI:** 10.3390/biomedicines12020302

**Published:** 2024-01-28

**Authors:** Aura Vîjîiac, Alina Ioana Scărlătescu, Ioana Gabriela Petre, Cristian Vîjîiac, Radu Gabriel Vătășescu

**Affiliations:** 1Cardiology Department, Carol Davila University of Medicine and Pharmacy, 050474 Bucharest, Romania; i_comanescu@yahoo.com (I.G.P.); radu_vatasescu@yahoo.com (R.G.V.); 2Emergency Clinical Hospital Bucharest, 014461 Bucharest, Romania; alina.scarlatescu@gmail.com; 3Emergency County Hospital Ploiești, 100097 Ploiești, Romania; cristivijiiac@yahoo.com

**Keywords:** dilated cardiomyopathy, atrioventricular coupling, three-dimensional echocardiography, combined atrioventricular coupling index

## Abstract

Atrioventricular coupling has recently emerged as an outcome predictor. Our aim was to assess, through three-dimensional (3D) echocardiography, the role of the left atrioventricular coupling index (LACI), right atrioventricular coupling index (RACI) and a novel combined atrioventricular coupling index (CACI) in a cohort of patients with dilated cardiomyopathy (DCM). One hundred twenty-one consecutive patients with DCM underwent comprehensive 3D echocardiographic acquisitions. LACI was defined as the ratio between left atrial and left ventricular 3D end-diastolic volumes. RACI was defined as the ratio between right atrial and right ventricular 3D end-diastolic volumes. CACI was defined as the sum of LACI and RACI. Patients were prospectively followed for death, heart transplant, nonfatal cardiac arrest and hospitalization for heart failure. Fifty-five patients reached the endpoint. All three coupling indices were significantly more impaired in patients with events, with CACI showing the highest area under the curve (AUC = 0.66, *p* = 0.003). All three indices were independent outcome predictors when tested in multivariable Cox regression (HR = 2.62, *p* = 0.01 for LACI; HR = 2.58, *p* = 0.004 for RACI; HR = 2.37, *p* = 0.01 for CACI), but only CACI showed an incremental prognostic power over traditional risk factors such as age, left ventricular strain, right ventricular strain and mitral regurgitation severity (likelihood ratio χ^2^ test = 28.2, *p* = 0.03). CACI assessed through 3D echocardiography, reflecting both left and right atrioventricular coupling, is an independent predictor of adverse events in DCM, yielding an incremental prognostic power over traditional risk factors.

## 1. Introduction

Dilated cardiomyopathy (DCM) is a heterogeneous disease that still causes substantial morbidity and mortality worldwide [1]. Since it was mainly considered a disease of the left ventricle (LV), most of the research focused on the evaluation of LV function, with the LV ejection fraction (LVEF) having a well-established prognostic role in DCM [2,3]. However, ventricular and atrial dysfunction are closely intertwined. Left atrial (LA) volume and strain are not only markers of LV diastolic dysfunction but also independent outcome predictors in heart failure (HF) [4,5]. Moreover, recent data have shown that both right ventricular (RV) and right atrial (RA) dysfunction have an independent prognostic role in left-sided HF [6,7,8].

The physiological interplay between the atrium and the ventricle suggests that the assessment of atrioventricular coupling could better reveal myocardial dysfunction and improve the outcome prediction [9]. A novel cardiac magnetic resonance (CMR)-derived left atrioventricular coupling index (LACI), defined as the ratio between LA end-diastolic volume (EDV) and LV EDV, proved to be an independent predictor of adverse events in patients without cardiovascular disease [9] and in patients with myocardial infarction [10]. Although three-dimensional (3D) LV and LA volumes showed a good correlation with CMR measurements [11], LACI has not been assessed so far using 3D echocardiography, and its utility in DCM is unknown. There is no data regarding the 3D right atrioventricular coupling index (RACI), defined as the ratio between RA EDV and RV EDV [12]. Therefore, the aim of our study was to assess, through 3D echocardiography, the prognostic role of LACI, RACI and a novel combined atrioventricular coupling index (CACI) in patients with DCM.

## 2. Materials and Methods

### 2.1. Study Population

The study population consisted of consecutive adult patients with DCM referred to our echocardiography laboratory between January 2019 and December 2021. Inclusion criteria were as follows: a dilated LV, according to cutoffs proposed in the current chamber quantification guidelines [13], with LVEF < 40%, in the absence of significant coronary artery disease or severe valvular heart disease [14]. All patients provided written informed consent prior to enrollment. Exclusion criteria included poor acoustic window, inability to hold breath, persistent atrial fibrillation, cor pulmonale and life expectancy < 1 year. The study protocol was approved by the ethics committee of our hospital. 

### 2.2. Echocardiography

Comprehensive two-dimensional (2D) and 3D echocardiographic acquisitions were performed using a Vivid E95 (GE Vingmed, Horten, Norway) ultrasound machine equipped with a 2.5 MHz 2D matrix array transducer and a 4V probe. Images were obtained with the patient in the left-lateral decubitus position, according to current recommendations [15], and they were subsequently analyzed offline using dedicated software (EchoPAC BT 13). 

LVEF and 2D LV volumes were measured using Simpson’s biplane disk summation rule from apical views [13,15]. LA and RA 2D volumes were obtained from the apical four-chamber view using the area-length method, providing the maximal volume (LA V_max_ and RA V_max_) at ventricular end-systole, the minimal volume (LA V_min_ and RA V_min_) at ventricular end-diastole, and the preA volume (LA V_preA_ and RA V_preA_) at the peak of the P wave on ECG [13]. The mitral E/E’ ratio was measured using the average tissue Doppler diastolic velocity of the mitral annulus at septal and lateral annular sites. Mitral regurgitation (MR) severity was graded as mild, moderate or severe based on qualitative and semiquantitative criteria proposed in current guidelines [16]. We used the apical RV-focused view [15,17] to measure the tricuspid annular plane systolic excursion (TAPSE) and pulsed tissue Doppler systolic velocity of the tricuspid annulus (S wave). RA pressure was estimated using the inferior vena cava diameter and collapsibility from the subcostal view. Pulmonary artery systolic pressure (PASP) was estimated as the sum of RA pressure and the tricuspid regurgitation systolic gradient [17]. 

For strain analysis, we used 2D strain software (EchoPAC—Q Analysis package) using high frame rate acquisitions (50–70 frames per second). For the assessment of LV global longitudinal strain (GLS-LV), apical four-, two- and three-chamber views were acquired, and the software automatically traced the endocardial border, which was manually readjusted when needed. The average peak systolic strain of all LV segments was reported as GLS-LV [18]. RV strain was calculated as the average of the peak systolic strain of the RV free wall segments (RVFW-LS). 

For 3D analysis, three experienced sonographers obtained six-beat full-volume 3D data sets with electrocardiographic gating during breath holding. For the left chambers, images were obtained from the apical four-chamber view, whereas for the right chambers, images were obtained from the apical RV-focused view. Datasets were analyzed offline using commercially available software: 4D Auto LVQ (EchoPAC v204, GE Vingmed, Horten, Norway) for LV analysis, 4D Auto RVQ (EchoPAC v204, GE Vingmed, Horten, Norway) for RV analysis and 4D Auto LAQ (EchoPAC v204, GE Vingmed, Horten, Norway) for LA and RA analysis. The software thus provided the biventricular EDV, ESV, EF, and the biatrial V_max_, V_min_ and V_preA_ at the same time in the cardiac cycle as the corresponding 2D volumes.

Based on the 3D atrial and ventricular volumes (Figure 1), we measured three atrioventricular coupling indices as follows: LACI, defined as the ratio between LA V_min_ and LV EDV; RACI, defined as the ratio between RA V_min_ and RV EDV; and CACI, defined as the sum between LACI and RACI. All the coupling indices were expressed as percentages, and a higher coupling index indicates a greater degree of atrioventricular decoupling.

3D acquisitions required an additional scanning time of 6 ± 3 min, while post-processing and offline analysis of biatrial and biventricular volumes required an additional 10 ± 4 min for each subject.

### 2.3. Follow-Up

Patients were followed up prospectively for the occurrence of any major adverse clinical event (MACE). The endpoint of this study was a composite of all-cause death, heart transplant, nonfatal cardiac arrest or readmission for heart failure exacerbation. For patients without MACE, we used the date of the last contact in survival analysis. Follow-up was conducted for a median of 19 ± 11 months. 

### 2.4. Statistical Analysis

Statistical analysis was performed using SPSS version 20.0 statistical software package and GraphPad Prism version 10.0.0 software. The Kolmogorov–Smirnov test was used to check for normal distribution. Continuous data were presented as mean ± standard deviation or as median and interquartile range, depending on the distribution. Categorical data were displayed as numbers and percentages. Comparison between continuous variables was conducted with Student’s *t*-test or Mann–Whitney U test, as dictated by distribution, while categorical variables were compared with Fisher exact test or χ^2^ test, as appropriate. 

Time-dependent receiver operating characteristic (ROC) curves and the respective area under the curve (AUC) were used to compare the ability of different parameters to predict the adverse outcome. Optimal cutoff values were chosen using the Youden index. Kaplan–Meier survival curves were compared using log-rank statistics. 

Atrioventricular coupling indices were tested in univariable and multivariable Cox proportional hazards regression. The multivariable model was constructed using variables with clinical relevance and/or statistical significance in univariable regression. To examine the incremental prognostic value of LACI, RACI and CACI over well-established predictors in DCM, we constructed nested models and compared them with the likelihood ratio χ^2^ test. A two-tailed *p*-value < 0.05 was considered statistically significant. 

We assessed the intra- and interobserver reproducibility of 3D measurements using intraclass coefficients (ICC) on a two-way mixed-effects model. Interobserver reproducibility was evaluated by having 10 datasets analyzed by two different researchers. Intraobserver reproducibility was tested using repeated measurements of 10 datasets two weeks apart, performed by the same researcher.

## 3. Results

After the exclusion of patients with persistent atrial fibrillation (15 patients), cor pulmonale (10 patients), severe clinical status with inability to hold breath (18 patients), poor acoustic window (7 patients) and life expectancy < 1 year (2 patients), 121 patients with DCM in sinus rhythm formed the final study population. The baseline clinical characteristics are summarized in Table 1. Most patients were male (74%), and most were in NYHA class II (59%), with a mean age of 59 ± 14 years. Seventy-eight patients (65%) had been diagnosed with DCM less than one year prior to enrollment. The etiology of DCM in the study group was heterogenous; there were 19 patients (16%) with familial cardiomyopathy, 21 patients (17%) with alcohol-induced cardiomyopathy, 13 patients (11%) with post-myocarditis DCM, 10 patients (8%) with tachycardiomyopathy, 7 patients (6%) with dyssynchrnopathy, 4 patients (3%) with chemotherapy-induced cardiomyopathy, 2 patients (2%) with LV noncompaction and 45 patients (37%) with idiopathic DCM.

Fifty-five patients (46%) experienced at least one MACE during follow-up; there were 26 deaths (22%), one heart transplant (1%), 5 nonfatal cardiac arrests (4%) and 23 readmissions for heart failure exacerbation (19%). Patients with events had a higher prevalence of NYHA class III and IV at enrollment (*p* < 0.001), a higher prevalence of diuretic use (*p* = 0.002) and higher brain natriuretic peptides (BNP) levels (*p* = 0.008). Except for diuretics, there were no differences regarding classes of HF drugs between patients with and without events (Table 1).

Table 2 presents 2D echocardiographic characteristics. There were no differences in LV systolic function between patients with and without events. However, patients with MACE had more severe LV diastolic dysfunction (*p* < 0.001 for mitral E/E’ ratio) and more impaired RV systolic function (*p* < 0.001 for TAPSE, S wave and RVFW-LS). All 2D LA and RA volumes were significantly larger in patients with events.

Table 3 presents 3D echocardiographic characteristics. There were excellent correlations between 2D and 3D LV volumes (r = 0.98, *p* < 0.001 for EDV; r = 0.99, *p* < 0.001 for ESV) and EF (r = 0.89, *p* < 0.001). All 2D LA volumes were positively correlated with the corresponding 3D volumes (r = 0.93, *p* < 0.001 for V_min_ and V_max_; r = 0.94, *p* < 0.001 for V_preA_). There were also excellent correlations between all 2D RA volumes and their corresponding 3D volumes (r = 0.96, *p* < 0.001 for V_max_; r = 0.97, *p* < 0.001 for V_preA_ and V_min_). All 3D LA and RA volumes were significantly larger in patients with MACE. While there were no differences in 3D LVEF between patients with and without events (*p* = 0.22), RVEF was significantly more impaired in patients with adverse outcomes (35 ± 8% versus 47 ± 7%, *p* < 0.001). All atrioventricular coupling indices were significantly larger in patients with MACE (*p* = 0.009 for LACI; *p* = 0.005 for RACI; *p* = 0.003 for CACI), reflecting a greater degree of decoupling. 

Good intra- and interobserver reproducibility were found for LACI (ICC = 0.94 [95% CI, 0.78–0.98] and ICC = 0.95 [95% CI, 0.81–0.99], respectively), RACI (ICC = 0.93 [95% CI, 0.75–0.98] and ICC = 0.85 [95% CI, 0.51–0.96], respectively) and CACI (ICC = 0.95 [95% CI, 0.83–0.99] and ICC = 0.91 [95% CI, 0.66–0.98], respectively).

In unadjusted Cox regression (Table 4), most of the 2D and 3D LA and RA volumes were significant predictors of MACE. All three atrioventricular coupling indices were significant predictors in univariable analysis (*p* = 0.005 for LACI; *p* = 0.03 for RACI; *p* = 0.006 for CACI). ROC analysis was used to compare the prognostic power of atrioventricular coupling indices (Table 5) and to find optimal cutoffs for event prediction based on the Youden index. CACI showed the best AUC (AUC = 0.66, *p* = 0.003), closely followed by RACI (AUC = 0.65, *p* = 0.005) and LACI (AUC = 0.64, *p* = 0.009). A CACI higher than 44% had a 78% sensitivity and a 52% specificity for MACE prediction. Kaplan–Meier curves showed a higher risk of events in patients with higher LACI, RACI and CACI (log-rank *p* = 0.001 for all) (Figure 2). 

All 3D atrial volumes and atrioventricular coupling indices were analyzed in multivariable Cox regression (Table 6). They were not tested together in the same model to avoid collinearity issues. The constructed model included well-established clinical and echocardiographic predictors in DCM (age, GLS-LV, MR severity and RVFW-LS) and, alternatively, one atrial volume or one atrioventricular coupling index at a time. We included GLS-LV and not LVEF in the model to avoid collinearity issues. None of the 3D LA volumes were independent predictors of MACE, while 3D RA V_min_ and RA V_preA_ remained independent outcome predictors. LACI, RACI and CACI were all independently associated with adverse events when tested as a categorical variable, but only CACI kept its prognostic power when tested as a continuous variable (Table 6). 

The likelihood ratio χ^2^ test demonstrated an incremental value of CACI (χ^2^ = 28.2, *p* = 0.03) to predict adverse outcomes on top of the multivariable model, including age, GLS-LV, MR severity and RVFW-LS (Figure 3). LACI and RACI yielded no incremental value (*p* = 0.07 and *p* = 0.08, respectively). 

## 4. Discussion

In this study, we found that 3D left and right atrioventricular coupling indices, as well as a novel 3D combined atrioventricular coupling index, were independent predictors of MACE in a cohort of patients with DCM. CACI showed the best association with adverse outcomes, having an incremental predictive value over traditional risk factors. 

The atrioventricular coupling index describes the hemodynamic relationship between the atrium and the ventricle in diastole [9]. The morphology and physiology of the LA reflect LV filling pressure and diastolic function [19,20]. While LA ESV is usually communicated in echo reports, since it is a well-known outcome predictor and marker of the chronicity of LV dysfunction [21,22], recent studies suggest that LA EDV is a better correlate of ventricular filling pressure and a better prognostic marker [4,23,24]. Moreover, a high LA EDV indicates an impaired atrial booster pump function. The change in atrial volume relative to that of the ventricle reflects the ventricular compliance, which in turn influences the atrial reservoir function, and LACI detects an earlier stage of atrial remodeling than atrial volumes alone [25]. Since it is based on the simultaneous measurement of LA and LV volume, LACI is supposed to integrate more accurately the atrial and ventricular performance [10], and it was proven to detect patients with heart failure and preserved LVEF with high diagnostic accuracy [26].

Based on these pathophysiological considerations, Pezel et al. investigated the role of CMR-derived LACI in a large cohort of patients free from cardiovascular disease at enrollment and found that this parameter was a strong predictor of incident heart failure [25] and major cardiovascular events [9]. CMR-derived LACI was also recently investigated in patients with cardiovascular disease, and it proved to be an independent predictor of adverse outcomes in patients with abnormal vasodilator stress CMR [27] and in patients following acute myocardial infarction, particularly in high-risk patients with LVEF <35% [10]. LACI was also assessed with cardiac CT, and it proved to be an independent predictor of cardiac death in a healthy population [28]. Meucci et al. recently found that LACI measured by 2D echocardiography was a strong predictor of atrial fibrillation in patients with hypertrophic cardiomyopathy [29]. However, 2D LA volumetric assessment is less accurate than 3D assessment because the atrium does not dilate uniformly; 3D echocardiography provides more precise measurements, free from geometric assumptions or atrial foreshortening [30]. Despite these advantages of 3D over 2D echocardiography, our study is the first assessment of atrioventricular coupling using 3D echocardiography. So far, the only other evaluation of RACI was undertaken using CMR in a cohort of pediatric patients with repaired tetralogy of Fallot, and it showed that CMR-derived RACI was a predictor of malignant arrhythmias [12].

Our study is the first evaluation of atrioventricular coupling in patients with DCM. The optimal LACI cutoff for event prediction was 20%, lower than the cutoffs found in the abovementioned studies (most of which were, however, based on CMR), which ranged from 25% to 40% [9,10,29]. The most likely explanation is the fact that DCM is a disease involving primarily the LV, leading to important ventricular dilation, often disproportionately when compared to the dilation of the LA. Therefore, LACI in DCM will fall within a lower range than in other cardiac diseases. More studies are needed to establish reference ranges for LACI and RACI across the spectrum of cardiac pathologies. So far, our study is the first evaluation of both LACI and RACI in the same cohort. 

The LA dilation and functional impairment in DCM are adaptative changes in response to LV dysfunction, but they may be partly explained using intrinsic atrial cardiomyopathy. The same is true for the RV, which undergoes morpho-functional changes secondary to afterload increase but which can also be affected by the same myopathic process as the LV. These changes in the RV will, in turn, lead to an impairment of RA function. In our study, LVEDV and RVEDV did not differ significantly between patients with and without MACE; however, LA and RA end-diastolic volumes (2D and 3D) were significantly larger in patients with events, suggesting different atrial adaptation patterns in response to impaired ventricular function. However, atrial volumes did not remain independent predictors in multivariable regression, while atrioventricular coupling indices did. The novel CACI, which was first described and investigated in the current study, showed the highest incremental predictive value when added over traditional risk factors. Furthermore, it was the only coupling index that retained its prognostic power when tested both as a categorical and as a continuous variable. This index considers changes in all four heart chambers; this can partly explain its prognostic value in our study since DCM is a disease that alters the morphology and function of all heart chambers. However, this is the first evaluation of CACI, and its predictive accuracy must be validated in larger external cohorts.

### 4.1. Clinical Implications

Since atrioventricular coupling proved to have better predictive value than atrial and ventricular volumes taken separately, early detection of atrioventricular mismatch should improve risk stratification in DCM. For this purpose, the novel CACI is probably the best choice because it includes both left and right atrioventricular decoupling. Since it is a unitless parameter, it should be easily reproducible among different 3D vendors, but further studies are needed. While 3D echocardiography is not yet routinely performed for all DCM patients, it did become more available and utilized during the last decade, and an advantage of CACI is that its measurement is based on volumes that are routinely reported in a 3D study. 

### 4.2. Study Limitations

While having the advantage of its prospective design, our study should be interpreted in the context of several limitations. It is a single-center experience with a small sample size and a relatively short follow-up period. Selection bias cannot be fully excluded since we did not include patients with atrial fibrillation to avoid stitch artifacts in 3D acquisitions. Moreover, there are no reference values in the general population for LACI, RACI and CACI, and the cutoffs derived from our ROC analysis are most likely not applicable to other cardiac diseases. Last but not least, the prognostic power of CACI was rather modest, and its potential impact on clinical decision-making remains to be determined.

## 5. Conclusions

Greater CACI measured with 3D echocardiography, indicative of left and right atrioventricular mismatch, was an independent predictor of major cardiovascular events in patients with DCM, showing an incremental prognostic power over traditional risk factors.

## Figures and Tables

**Figure 1 biomedicines-12-00302-f001:**
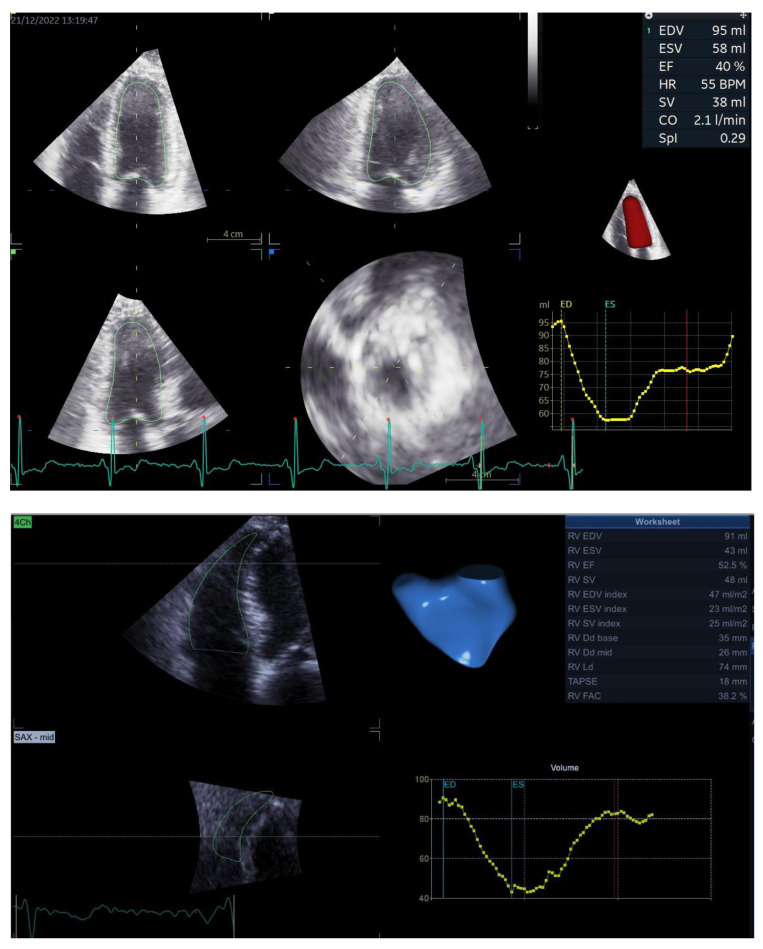
3D assessment using dedicated software for analysis of LV volumes and EF (**upper panel**), RV volumes and EF (**middle panel**) and atrial volumes (**lower panel**), upon which measurements of LACI, RACI and CACI are based. Abbreviations are in the text.

**Figure 2 biomedicines-12-00302-f002:**
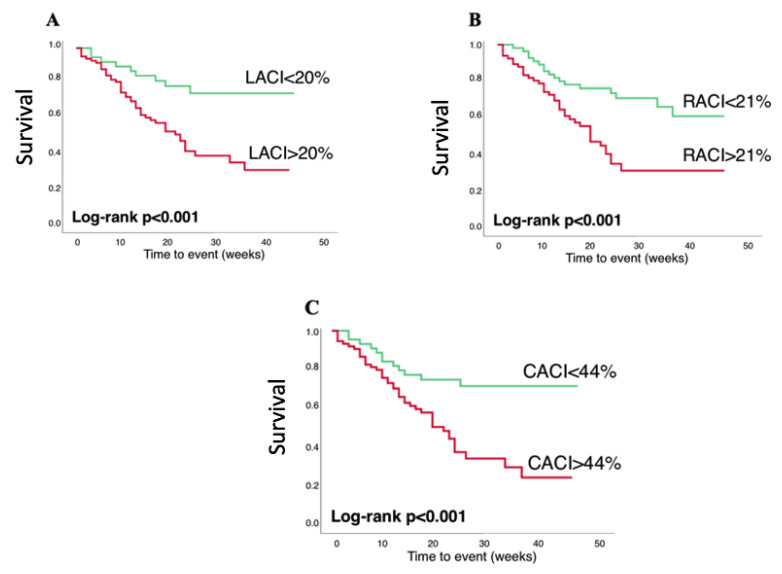
Kaplan–Meier survival curves stratified using left atrioventricular coupling index (**A**), right atrioventricular coupling index (**B**) and combined atrioventricular coupling index (**C**).

**Figure 3 biomedicines-12-00302-f003:**
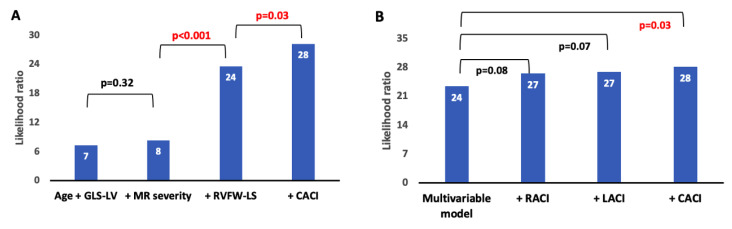
Bar charts showing the incremental prognostic value of atrioventricular coupling indices over traditional prognostic factors presented as a likelihood ratio χ^2^ test. (**A**) The addition of RVFW-LS and subsequently of CACI over nested models, including age, GLS-LV and MR severity, led to significant incremental prognostic power. (**B**) The addition of CACI, but neither LACI nor RACI, yielded an incremental prognostic value over the multivariable model, including age, GLS-LV, MR severity and RVFW-LS.

**Table 1 biomedicines-12-00302-t001:** Baseline clinical characteristics.

Variables	All Patients (n = 121)	MACE (n = 55)	No MACE (n = 66)	*p*
Age (years)	59 ± 14	59 ± 15	59 ± 14	0.92
Male, n (%)	89 (74%)	40 (73%)	49 (74%)	0.85
Comorbidities, n (%)				
Diabetes	29 (24%)	15 (27%)	14 (21%)	0.44
Hypertension	81 (6%)	33 (60%)	48 (73%)	0.14
Smoking	44 (36%)	20 (36%)	24 (36%)	1.00
Systolic BP (mm Hg)	125 ± 19	122 ± 19	128 ± 18	**0.047**
Diastolic BP (mm Hg)	80 (65–80)	70 (60–80)	80 (70–85)	**0.001**
Heart rate (bpm)	81 ± 16	82 ± 16	79 ± 15	0.35
NYHA class, n (%)				**<0.001**
II	71 (59%)	22 (40%)	49 (74%)	
III	43 (35%)	27 (49%)	16 (24%)	
IV	7 (6%)	6 (11%)	1 (2%)	
Medication, n (%)				
ACE-I/ARBs/ARN-I	115 (95%)	52 (95%)	63 (95%)	0.82
β blockers	117 (97%)	51 (93%)	66 (100%)	**0.04**
MRA	108 (89%)	51 (93%)	57 (86%)	0.26
SGLT2-I	17 (14%)	7 (13%)	10 (15%)	0.70
Loop diuretics	79 (65%)	44 (80%)	35 (53%)	**0.002**
NT pro-BNP (pg/mL)	480 (163–1525)	690 (211–1815)	253 (105–505)	**0.008**

Units of measurement are in parentheses. Abbreviations are in the text. Bolded *p* values are statistically significant.

**Table 2 biomedicines-12-00302-t002:** 2D echocardiographic characteristics.

Variables	All Patients (n = 121)	MACE (n = 55)	No MACE (n = 66)	*p*
2D LVEDV index (mL/m^2^)	125 ± 45	132 ± 49	118 ± 40	0.10
2D LVESV index (mL/m^2^)	95 ± 40	102 ± 44	89 ± 36	0.07
2D LVEF (%)	25 ± 7	24 ± 7	26 ± 7	0.08
Mitral E/E’ ratio	16 ± 7	18 ± 8	13 ± 5	**<0.001**
MR severity, n (%)				0.06
Mild	87 (72%)	34 (62%)	53 (80%)	
Moderate	26 (21%)	17 (31%)	9 (14%)	
Severe	8 (7%)	4 (7%)	4 (6%)	
GLS-LV (%)	−6.8 ± 2.8	−6.3 ± 3.1	−7.2 ± 2.5	0.09
2D LA V_max_ index (mL/m^2^)	52 ± 25	61 ± 25	45 ± 22	**<0.001**
2D LA V_min_ index (mL/m^2^)	35 ± 20	42 ± 20	30 ± 18	**<0.001**
2D LA V_preA_ index (mL/m^2^)	43 ± 21	50 ± 21	37 ± 20	**<0.001**
RV basal diameter (mm)	39 ± 8	41 ± 8	38 ± 7	0.08
TAPSE (mm)	17 (15–20)	16 (12–18)	19 (17–23)	**<0.001**
S wave (cm/s)	11 ± 3	9 ± 2	12 ± 2	**<0.001**
2D RA V_max_ index (mL/m^2^)	29 (20–40)	34 (21–46)	25 (18–37)	**0.02**
2D RA V_min_ index (mL/m^2^)	17 (11–27)	21 (12–31)	15 (10–23)	**0.02**
2D RA V_preA_ index (mL/m^2^)	23 (16–34)	28 (17–38)	19 (15–31)	**0.009**
PASP (mm Hg)	41 ± 17	45 ± 18	38 ± 16	**0.04**
RVFW-LS (%)	−13.9 ± 8.1	−11.0 ± 8.4	−16.4 ± 7.1	**<0.001**

Units of measurement are in parentheses. Abbreviations are in the text. Bolded *p* values are statistically significant.

**Table 3 biomedicines-12-00302-t003:** 3D echocardiographic characteristics.

Variables	All Patients (n = 121)	MACE (n = 55)	No MACE (n = 66)	*p*
3D LVEDV index (mL/m^2^)	133 ± 48	142 ± 52	124 ± 43	**0.046**
3D LVESV index (mL/m^2^)	100 ± 43	109 ± 46	93 ± 38	**0.047**
3D LVEF (%)	26 ± 7	25 ± 7	26 ± 7	0.22
3D LA V_max_ index (mL/m^2^)	54 (39–66)	60 (51–72)	47 (31–59)	**<0.001**
3D LA V_min_ index (mL/m^2^)	38 ± 20	44 ± 20	33 ± 19	**0.002**
3D LA V_preA_ index (mL/m^2^)	44 (33–56)	50 (41–61)	36 (24–51)	**<0.001**
3D RVEDV index (mL/m^2^)	83 ± 32	85 ± 35	81 ± 29	0.44
3D RVESV index (mL/m^2^)	48 ± 20	55 ± 23	43 ± 16	**0.001**
3D RVEF (%)	42 ± 9	35 ± 8	47 ± 7	**<0.001**
3D RA V_max_ index (mL/m^2^)	35 ± 20	41 ± 24	30 ± 14	**0.007**
3D RA V_min_ index (mL/m^2^)	18 (12–29)	24 (15–34)	16 (11–25)	**0.003**
3D RA V_preA_ index (mL/m^2^)	25 (17–36)	31 (20–42)	21 (14–32)	**0.004**
LACI (%)	26 (17–40)	29 (22–41)	22 (16–36)	**0.009**
RACI (%)	23 (16–40)	28 (19–45)	19 (15–30)	**0.005**
CACI (%)	51 (36–80)	59 (44–97)	42 (34–61)	**0.003**

Units of measurement are in parentheses. Abbreviations are in the text. Bolded *p* values are statistically significant.

**Table 4 biomedicines-12-00302-t004:** Unadjusted Cox regression for MACE.

Variables	HR (95% CI)	*p*
2D LVEDV	1.00 (1.00–1.01)	0.09
2D LVESV	1.00 (1.00–1.01)	**0.04**
2D LVEF	0.95 (0.92–0.99)	**0.01**
GLS-LV	1.15 (1.04–1.27)	**0.009**
RVFW-LS	1.08 (1.05–1.12)	**<0.001**
2D LA V_max_	1.01 (1.00–1.01)	**0.001**
2D LA V_min_	1.01 (1.00–1.02)	**0.001**
2D LA V_preA_	1.01 (1.00–1.01)	**0.001**
2D RA V_max_	1.01 (1.00–1.01)	**0.004**
2D RA V_min_	1.01 (1.00–1.02)	**0.003**
2D RA V_preA_	1.01 (1.00–1.01)	**0.002**
3D LVEDV	1.00 (1.00–1.01)	**0.04**
3D LVESV	1.00 (1.00–1.01)	**0.03**
3D LVEF	0.97 (0.93–1.00)	0.08
3D RVEDV	1.00 (1.00–1.01)	0.30
3D RVESV	1.01 (1.01–1.02)	**<0.001**
3D RVEF	0.87 (0.84–0.90)	**<0.001**
3D LA V_max_	1.01 (1.00–1.01)	**<0.001**
3D LA V_min_	1.01 (1.00–1.02)	**0.001**
3D LA V_preA_	1.01 (1.00–1.02)	**<0.001**
3D RA V_max_	1.01 (1.00–1.01)	**0.003**
3D RA V_min_	1.01 (1.00–1.02)	**0.002**
3D RA V_preA_	1.01 (1.00–1.02)	**0.001**
LACI	1.02 (1.01–1.04)	**0.005**
LACI > 20%	3.21 (1.57–6.59)	**0.001**
RACI	1.01 (1.00–1.02)	**0.03**
RACI > 21%	2.63 (1.47–4.68)	**0.001**
CACI	1.01 (1.00–1.02)	**0.005**
CACI > 44%	2.91 (1.53–5.55)	**0.001**

HR—hazard ratio; CI—confidence interval. The rest of the abbreviations are in the text. Bolded *p* values are statistically significant.

**Table 5 biomedicines-12-00302-t005:** ROC analysis for MACE prediction.

Variables	AUC (95% CI)	*p*	Cut-Off	Sensitivity	Specificity
LACI	0.64 (0.54–0.74)	0.009	20%	84%	44%
RACI	0.65 (0.55–0.75)	0.005	21%	71%	58%
CACI	0.66 (0.56–0.76)	0.003	44%	78%	52%

AUC—area under the curve; CI—confidence interval. The rest of the abbreviations are in the text.

**Table 6 biomedicines-12-00302-t006:** Adjusted * Cox regression for MACE.

Variables	HR (95% CI)	*p*
3D LA V_max_	1.01 (1.00–1.01)	0.06
3D LA V_min_	1.01 (1.00–1.01)	0.14
3D LA V_preA_	1.01 (1.00–1.01)	0.08
3D RA V_max_	1.01 (1.00–1.01)	0.06
3D RA V_min_	1.01 (1.00–1.02)	**0.046**
3D RA V_preA_	1.01 (1.00–1.02)	**0.03**
CACI > 44% (yes/no)	2.37 (1.20–4.70)	**0.01**
CACI (per unit increase)	1.01 (1.00–1.02)	**0.03**
LACI > 20% (yes/no)	2.62 (1.22–5.60)	**0.01**
LACI (per unit increase)	1.02 (1.00–1.04)	0.07
RACI > 21% (yes/no)	2.58 (1.36–4.90)	**0.004**
RACI (per unit increase)	1.01 (1.00–1.03)	0.09

* Adjusted for age, GLS-LV, MR severity and RVFW-LS. HR—hazard ratio; CI—confidence interval. The rest of the abbreviations are in the text. Bolded *p* values are statistically significant.

## Data Availability

Data are available upon request.

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
