# Peer review of "Three-Dimensional Combined Atrioventricular Coupling Index—A Novel Prognostic Marker in Dilated Cardiomyopathy"

_biomedicines, 2024, doi:10.3390/biomedicines12020302_

Round 1

Reviewer 1 Report

Comments and Suggestions for Authors

Congratulations to the authors. Very interesting analysis and results.

I have only one comment:

It would be advisable to illustrate the LACI, RACI and CACI measurements performed

Author Response

We addressed the reviewer's suggestion in Figure 1. Thank you

Reviewer 2 Report

Comments and Suggestions for Authors

Thank you to the authors for this well-conducted and interesting study for the community of cardiologists.

However, I have 2 critiques:

In the methods It is necessary to describe how much additional time is necessary to obtain the 3D data, distinguishing between additional time close to the patient and post-processing of the data.

The intra and inter-observer reproducibility of the 3D measurements is missing. This methodological error must be corrected.

Author Response

We addressed the reviewer's comments in the "methods" and "results" section, respectively. Thank you.

Reviewer 3 Report

Comments and Suggestions for Authors

1. The aim of the study was to assess the prognostic role of LACI, RACI  (defined as the ratio between atrial and ventricular EDV) and a novel combined atrioventricular coupling index (CACI, the sum of the two) by 3D echocardiography in patients  with DCM.

Models for prognostic risk prediction are widely used in the cardiovascular field to predict risk of future events or to stratify apparently healthy individuals and patients into risk categories, even more when supposedly promising risk markers are identified whose validation is an important task. The validation of prognostic models requires calibration, external validation (two topics not addressed in this study) and discrimination, this latter estimated by

-C statistic, a function of the sensitivity and specificity of the indices under validation

-Likelihood ratio(LR)s, the ratio between the probability that LACI, RACI and CACI would correctly identify a different prognostic outcome according to a certain threshold.

However, both statistics performed well below the standards expected to allow effective discrimination. In fact, a C-statistics of 0.66 or so allows a correct prediction of an event at expense of a about one third of false positive results and certainly would be much, much less when taking into account preliminarily other well recognized predictors. As regards LRs both RACI and LACI did not increment the discrimination power in a multivariate model while the contribution of CACI, albeit somewhat statistically significant, was quantitatively modest and indeed not sufficient to change drastically the process of decision making, as you may want from a clinically relevant marker.

 2.The closest correlation between 2D and 3D echocardiographic measurements makes 3D a rather superfluous surrogate of what 2D measurements can obtain.

 3. The endpoint of this study was a composite of all-cause death, heart transplant, nonfatal cardiac arrest, or readmission for heart failure exacerbation but a more detailed specification of the clinical outcomes would be more useful to evaluate the relationship between the atrioventricular coupling indices and the clinical evolution of this sample of DCM pts

 4. Is it pathophysiologically sound to derive a prognostic index by simply summing its left and right  components?

Author Response

  1. Indeed, the prognostic value of coupling indices as assessed by AUC was satisfactory, not excellent. There are several well-established prognostic markers in DCM which are widely used; however, in multivariable regression, the model including CACI did show an incremental predictive value over such well-established predictors. Since this is the first definition and evaluation of CACI, we believe that our study opens the way for further investigation of this parameter, not only in DCM, but also in other cardiac diseases, in order to establish whether its prognostic power is clinically relevant in daily practice. We modified the "limitations" section accordingly.
  2. While 2D and 3D measurements showed very good correlation, it has been established in the last decade that 3D volumes are better correlated with the gold standard represented by magnetic resonance, and that they are better outcome predictors than 2D volumes. Moreover, while LV volumes (and thus LACI) can be measured with 2D echo, RV volumes (and thus RACI, and CACI) can only be measured with 3D echo. 
  3. Although the study had a composite endpoint, we analyzed if atrioventricular coupling indices were better predictors of death or of rehospitalization (since nonfatal cardiac arrests and heart transplant had a low prevalence in our study). None of the coupling indices predicted rehospitalization. Both LACI, RACI and CACI were predictors of death, with AUC between 0.6 and 0.7; this is very similar to the results using the composite endpoint, hence analyzing death separately didn't bring any additional information and it felt redundant to put it in the "results" section.
  4. In left heart disease the prognostic becomes worse when right heart failure ensues (and vice versa, in congenital heart disease primarily involving the right heart, the development of LV dysfunction worsens the prognostic). Hence, we hypothesised that a parameter which incorporates both left heart dysfunction and right heart dysfunction will be a better risk predictor than parameters of left or right failure taken separately. 

Round 2

Reviewer 2 Report

Comments and Suggestions for Authors

thank you for providing appropriate responses to my criticisms

Author Response

Thank you

Reviewer 3 Report

Comments and Suggestions for Authors

From a statistical point of view, a C-statistics of 0.66 for LACI implies to accept a trade-off between 84% of true positives and 56% of false positives and as regards RACI and CACI the corresponding figures are 71% vs 41% and 78% vs 48% respectively (see Table 5) and from my perspective, neither LACI, RACI and CACI qualify as "satisfactory" prognostic markers. Moreover, had the Authors calculated C-statistics for LACI, RACI and CACI AFTER accounting for age, GLS-LV, MR severity and RVFW-LS as they have correctly  done for the likelihood ratios, C values would have most likely increased by just one or two units.

However, C-statistics are rather insensitive as a prognostic discrimination tool and likelihood ratios work better in that sense. Still LACI and RACI proved of no statistically significant benefit in that regard while the independent contribution of CACI (the arithmetic addition of LACI and CACI whose pathophysiological meaning leaves me uncertain while certainly it needs further validation) albeit significant was quantitatively rather immaterial  according to the present results (about 5% incremental power as compared to RACI and LACI considered alone: from 26.9 and 26.5 to 28.2). Even more since the more advanced disease severity of those patients with pending MACE could be easily predicted on the basis of the clinical parameters reported in Table 1.

The above considerations do not want to slight the relevant and easily recognizable echocardiographic skills of the Authors but these limitations should have been clearly discussed (that were'nt) in a document whose primary aim was to assess the PROGNOSTIC role of a purported novel and potentially clinically useful prognosticator.

Author Response

Although we agree that the incremental predictive power of CACI was rather modest, we do consider LACI, RACI and CACI "satisfactory" prognostic markers, since this is the term used in statistics to characterize the predictive accuracy of an AUC between 0.6 and 0.7. While the results of our study would probably be unworthy for a scrupulous statistician, this is a pilot, hypothesis-generating, single-center study, and external validation is necessary to establish the real-world value of these parameters. In fact, we are currently undergoing evaluation of 3D coupling indices in different cardiac diseases in partnership with other 2 tertiary centers, but this is beyond the scope of the present study. Our results are based on the data we gathered and analyzed so far. Considering we cannot change the data we have, we don't fully understand what further changes the reviewer would want us to make in our manuscript. Thank you